# Patient and clinician perceptions of telehealth in musculoskeletal physiotherapy services - A systematic review of the evidence-base

Anthony Smith [ID]*, Sue Innes [ID]

School of Rehabilitation and Exercise Sciences, University of Essex, Colchester, United Kingdom

* apsmitb@essex.ac.uk

## Abstract

Telehealth has been at the forefront of healthcare delivery since the Covid-19 pandemic with a prompt shift in transition from face-to-face delivery to remote contact. This critical review aims to understand patient and clinician views of telehealth adoption regarding effectiveness and satisfaction within musculoskeletal (MSK) physiotherapy services. A systematic process was used to search for evidence within 6 databases (CINAHL, PyscINFO, Medline, AMED, EMCARE, EMBASE) utilising clear inclusion and exclusion criteria in August 2024. Articles published in English between 2019-2024 were searched, a total of 394 articles were identified and 10 articles were included in the review. Methodological quality was evaluated using the CASP, JBI and QuADS tools. Findings were evaluated via consensus and showed clear patient and clinician satisfaction with positive themes of reduced travel, reduced physical burden, flexibility/accessibility and negative themes of reduced physical contact, computer literacy and privacy infringements. Quality analysis identified non-response bias, sampling bias, and participants mix as risks to overall validity. Telehealth has shown to be an effective and transformative model of healthcare delivery for musculoskeletal services, especially in improving access and convenience for patients. Implications for practice suggest a need for a hybrid model of care, enhanced training, and improved data security. Future research should focus on satisfaction within condition specific musculoskeletal health, overall cost-effectiveness, health equity, and the integration of advanced technologies to ensure telehealth can be a sustainable and inclusive part of the healthcare landscape moving forwards.

## Author summary

On the 11th of March 2020, the World Health Organisation (WHO) declared a global health pandemic because of rapid transmission of the Covid-19 virus. The impacts of Covid-19 on healthcare systems caused a prompt evaluation of healthcare service provision to supress transmission rates and optimise emergency treatment of Covid-19 patients. As a result of this, almost all non-essential physiotherapy consultations were

**Data availability statement:** All relevant data are within the manuscript and its Supporting information files.

**Funding:** The author(s) received no specific funding for this work.

**Competing interests:** The authors have declared that no competing interests exist.

cancelled, and guidance given for virtual remote consultation and triage. Telehealth consultations increased exponentially as a direct result of Covid-19 and telehealth has continued to form an integrated part of health care services. Both the Royal College of Physicians, and the Chartered Society of Physiotherapy have produced guidelines for this to be embedded into future service provision. The aim of this review is to understand the perceptions of both patients and clinicians towards telehealth.

## Introduction

Musculoskeletal (MSK) physiotherapists working in both primary and secondary care roles provide expert assessment, treatment, and management of MSK conditions. Physiotherapy improves human movement and function to optimise health in individuals affected by illness, injury, or disability [1]. MSK physiotherapy as a healthcare profession has traditionally used face-to-face consultation to gather information through history taking and physical examination, this assessment then enables decision making and treatment. Historically, General Practitioners (GP's) and other Allied Health Professionals (AHP) refer to MSK physiotherapy services after initial consultation, however there is a growing number of MSK physiotherapists working within primary care offering initial consultation and management as a first point of contact [2].

Telemedicine is synonymous with other terms such as telehealth, telerehabilitation, tele-physiotherapy, teleconsultation and telediagnosis to name but a few [3–7]. Telehealth service provision is not a new concept to MSK physiotherapy, and numerous systematic reviews demonstrate efficacy in a wide variety of MSK conditions [8,9]. Historically there has been slow uptake and implementation of telehealth prior to the Covid-19 pandemic with most telehealth services enlisted to improve healthcare access in remote locations or in the event of reduced workforce capacity [10]. The wider role out of telehealth services within physiotherapy has been restricted by local technology infrastructure and cost implications of up scaling Information Technology (IT) systems. The lack of reimbursement from funding bodies for upgrading hardware and software was likely conducive to limited uptake prior to the pandemic [11]. However, since the Covid-19 pandemic there has been a shift in policy regarding funding of this utility [12] and combined with the need to access healthcare in a safe and effective manner during the Covid-19 pandemic has likely added to the surge of telehealth services across a variety of healthcare settings including MSK physiotherapy.

The Royal College of General Practitioners [13] published guidance about the future potential of remote consultation and have forecast plans to embed this utility of service delivery firmly in healthcare within a primary care setting. This potentiates implications for physiotherapists working within primary care settings such as first point of contact clinicians as they will likely adopt a hybrid approach to patient assessment and access to healthcare.

Consideration should be given to potential barriers to telehealth implementation and whether this could de-rail plans to embed this into routine MSK practice. The rapid expansion of telehealth and virtual services over such a short period of time may have resulted in sub-optimal healthcare delivery and unduly discriminate against hard-to-reach patient groups such as the elderly, disabled or patients without access to appropriate technology [11,14,15]. Understanding clinician barriers to implementation could equally affect the outcome of successful integration into routine healthcare models. Barriers such as technology literacy, resistance to workplace change, de-personalisation of care, safeguarding and privacy concerns have been highlighted as potential obstacles [10].

The aim of this critical review will be to identify and consider research that will highlight patient and clinician satisfaction with telehealth service delivery within musculoskeletal physiotherapy services and to discuss how this may impact on future integration within routine clinical practice.

## Formulation of research question

The PEO format was used to develop a specific research question and has been described in the literature by various authors when developing specific qualitative research questions [16–18].

The proposed research question is:

*"In musculoskeletal physiotherapy, what impact has telehealth provision had on patient and clinician satisfaction?"*

The key research aim of this critical review is to understand patient and clinician views of telehealth adoption within musculoskeletal physiotherapy services.

## Methods

### Trial registration

This review was registered in the PROSPERO database: CRD42024498878.

**Search of literature.** Medical Search Terms or Subject Headings (MeSH) were derived from thesaurus headings (S1 Panel) within each individually searched database (AMED, CINAHL, EMBASE, EMCARE, Medline and PsycINFO) and development of synonyms or exact phrases was undertaken to increase the breadth of search terms. Additional search techniques were added to increase the ability to capture variations of search terms. Truncation and wildcard techniques were used to find divergent search terms, finding singular and plural search terms, and variations of root word endings (S1 Panel). This allowed for simultaneous searching of multiple free text variations using a singular search term saving the researcher valuable time. This search strategy is all-inclusive and allows focused searching of the databases [19].

Searching of the databases was undertaken via OVID, EBSCO host and ProQuest platforms. A succinct search was undertaken of a total of 30 article titles within each database to form a list of free text search terms that could exclude non-relevant studies from the overall search strategy. A peer review of the selected search terms was undertaken, and a comprehensive list was identified to filter the remaining studies using the 'NOT' Boolean operator. Database limiters were added to filtrate and taper down the gathered articles into more relevant studies for the critical review.

Database searching produced a moderate amount of good quality articles. The decision was made to exclude grey literature, however, reference list and citation searching was undertaken [20] to explore any additional article sources and this has been highlighted within the PRISMA search.

**Selection of articles.** Each stage of the PRISMA process included a double-screened peer review whereby two researchers (AS and SI) independently screened and reviewed the literature at the title, abstract and full text stages. This process gained agreement that comprised studies met the inclusion criteria (Table 1). Any disagreements between reviewers on the inclusion/exclusion of articles were discussed fully and where agreement could not be reached a third independent review person was enlisted to assist with resolving any discrepancies of the included articles. It was decided that 100% agreement was needed

**Table 1. Inclusion/Exclusion Criteria.**

| | INCLUSION CRITERIA | EXCLUSION CRITERIA |
|---|---|---|
| Publication Type | • Peer reviewed journal articles<br>• Full Text Articles<br>• English Language Available | • Reviews, proposals, dissertations, non-peer reviewed documents, book chapters, poster presentations<br>• Non-Full Text Articles<br>• Non-English Language Studies |
| Study Type | • Primary Research<br>• Quantitative, Qualitative and Mixed Method | • Secondary Research<br>• Systematic Review/Meta-analysis |
| Populations | • Musculoskeletal Physiotherapy - Service Provision<br>• Adults (>18 years of age) with Musculo-skeletal Disorders | • Specific Musculoskeletal Physiotherapy Conditions<br>• Other Non-MSK Physiotherapy Services (Neurology, Respiratory, Community)<br>• Non-Physiotherapy Services (GP's, Consultant Physicians)<br>• Children/Young Adults (<18 years of age) |
| Exposure | • Telehealth Services as main Exposure/Intervention or part of stratified approach | • Non-Telehealth Exposures/Interventions |
| Outcomes | • Primary/secondary outcome of Service Satisfaction<br>• Patient/clinician Satisfaction (Attitudes, views, perceptions, experiences, acceptance) | • Satisfaction not included as an outcome measure. |
| Year of Publication | • Studies between 2019 – 2024 | • Studies before or after 2019 – 2024 |

between independent reviewers for inclusion of the specific study. Where agreement could not be reached by a third-party reviewer, the study was maintained for review and inclusion within the study to avoid missing relevant research.

Database searches yielded a total of 394 articles. No duplicate articles were identified. An additional 8 articles were included via reference list and citation searching. 402 articles remained which were further screened by study title. 290 articles were excluded after screening by title with most articles deemed irrelevant to the research question or excluded as systematic reviews. The second round of screening followed by review of abstracts with a further 92 articles excluded leaving the total remaining articles to be screened by full text review as 20. The reasons given for exclusion at the abstract stage were recorded (Fig 1), with the largest number of articles excluded as they did not meet the inclusion criteria. Screening of full text excluded another 10 studies (S2 Panel). The reason for exclusion included incorrect population, outcomes, and poster presentation. The remaining 10 studies were deemed to meet the inclusion/exclusion criteria and were included in the final review.

**Data extraction and analysis.** All included studies were published and available for full review. Extraction of data was undertaken on September 2024 by one reviewer (AS) and verified by the second reviewer (SI). The data extracted included details regarding study design, outcome measure, measurement tool, whether patient or clinician data collected, clinical setting, country of origin, sample size, gender percentages, mean age, and survey response rate. Study analysis was undertaken to form a narrative theme of findings for this critical review. The process of content thematic analysis has been used to identify patterns across the studies about perceptions of telehealth use and quality of literature that are deemed important and are associated to answering the specific research question and aims of this review. Narrative analysis of specific identified themes followed which was used to answer the

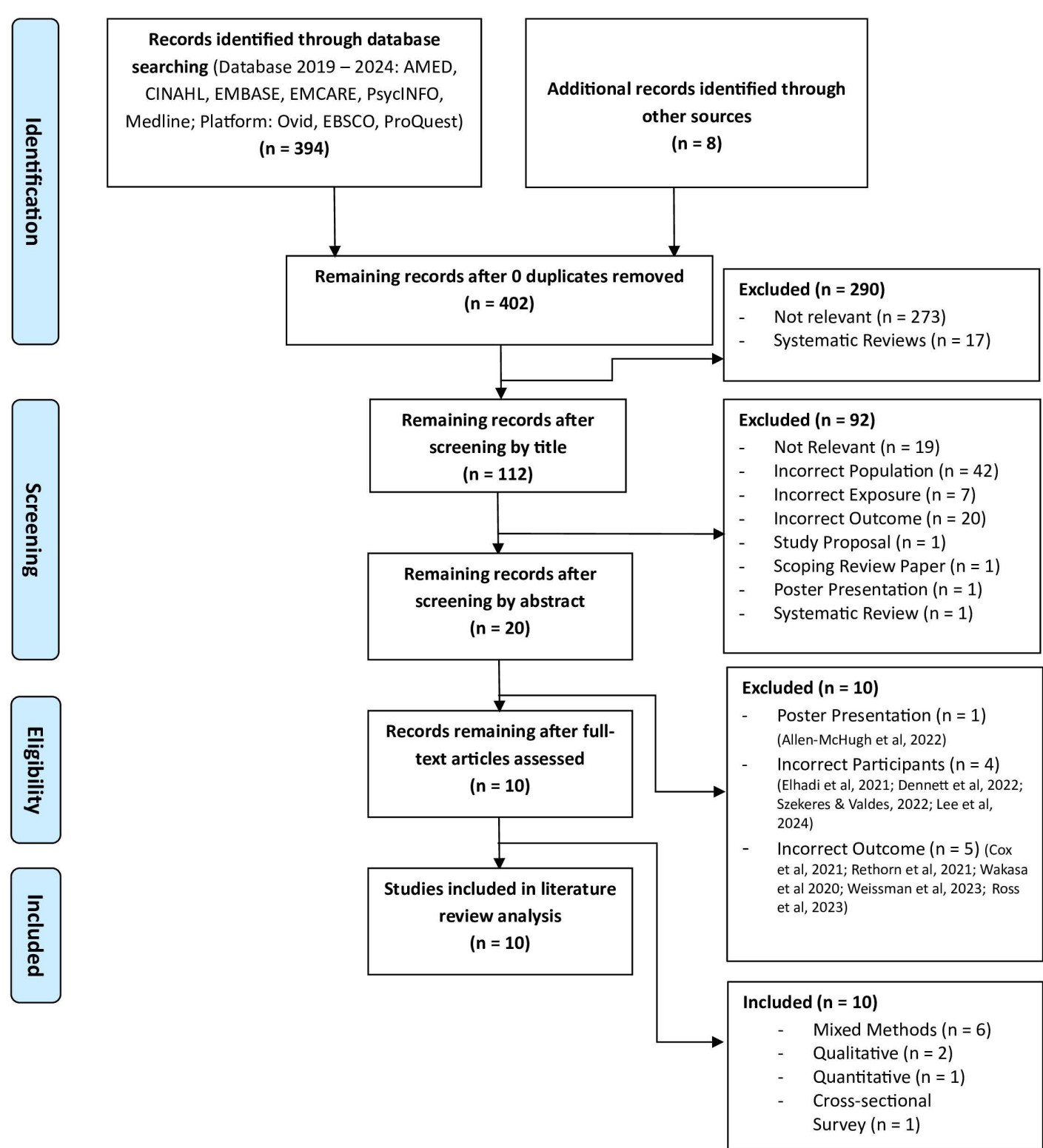

**Fig 1. PRISMA Flowchart.**

research question. The process of thematic analysis has been used extensively in qualitative research and offers researchers a method of analysis that is practicable [21,22].

## Results

### Characteristics of studies

A summary of results characteristics was formulated to identify key characteristics of the included studies. (Table 2).

### Key findings

Despite the synonymy between the different dimensions studied, firm conclusions between studies are limited by the heterogenous nature of methodology, however, a narrative exploring themes of satisfaction from the collected studies is possible.

**Table 2. Characteristics of studies.**

| Author | Sample | Country | Design | Outcome | Patient or Clinician | Male (%) | Age (y) Mean (SD) | Response Rate (%) |
|---|---|---|---|---|---|---|---|---|
| Malliaras et al. [23] | n = 1185 | Worldwide | Online Survey | Use and Views of Telehealth | 688 (82%) Physio | 392 (46.8%) | 38y ± 10.4 | 827 (69%) |
| Bennell et al. [24] | n = 638 | Australia | Online Survey | Implementation & experiences of Telehealth | 218 (34%) Physio 420 (66%) Patients | Physio 55 (27%) Patients 95 (24%) | Physio (Not collected) Patient Ordinal Data – no mean or SD | Physio 207 (95%) Patient 401 (95%) |
| Miller et al. [12] | n = 1501 | USA | Online Survey + Interview | Reach, Effectiveness, Adoption, Implementation, Maintenance | 307 (27%) Patients 19 Clinicians | 532 (35%) | Patients Ordinal Data – no mean or SD, no data for satisfaction survey participants | Patient 307 (27%) |
| Reynolds et al. [25] | n = 2587 | Ireland | Online Survey | Use and Views of Telehealth | Physio | 46 (22.44%) | Mean 36y | Physio 205 (8%) Sample size calculation of 193 |
| Albahrouh & Buabbas, [26] | n = 747 | Kuwait | Online survey + Semi-structured Interview | Perceptions and willingness to use Telehealth | Physio | 94 (34.4%) | 59% 35-50y | 273 (36.5%) |
| Jansen-Kosterink et al. [27] | n = 118 | Netherland | Qualitative Focus Group Discussion | Explore acceptance of Telehealth | Patient | N/A | N/A | N/A |
| Odole et al. [28] | n = 6 | Nigeria | Qualitative Focus Group Discussion | Explore perception of telehealth services | Physio | N/A | N/A | N/A |
| Ullah et al. [29] | n = 82 | Saudi Arabia | Online Survey | Explore knowledge and confidence with tele-rehabilitation | 22 (26.83%) Physio | 52 (63.41%) | No data | No data |
| Barton et al. [30] | n = 186 | Australia | Online Survey + Interview | Experiences and attitudes of patients to Telehealth | Patient | 65 (38%) | 49y ± 28 | 172 (93%) |
| Fernandes et al. [31] | n = 1107 | Brazil | Online Survey | Understand acceptability, preferences, and needs of Telehealth | 707 (64%) Physio 400 (36%) Patients | 200 (28.3%) Physio 139 (34.8%) Patients | 33.6y ± 7.6 Physio 34.2y ± 12.6 Patient | 90-100% for each item |

Studies consisted of mixed method (n = 6), qualitative (n = 2), quantitative (n = 1), and cross-sectional (n = 1) study design. Studies were conducted from a variety of geographical locations including Worldwide (n = 1), USA (n = 1), Australia/New Zealand (n = 2), Ireland (n= 1), Kuwait (n = 1), Netherlands (n = 1), Nigeria (n = 1), Brazil (n = 1), and Saudi Arabia (n = 1). Studies focused on clinician viewpoints (n = 5), patient viewpoints (n = 2), or both clinician/patient viewpoints (n = 3) of telehealth.

1. *Overall Satisfaction*

Bennell et al. [24] and Miller et al. [12] shared similar findings on patient satisfaction with a high percentage reporting a good response to telehealth (92%-94%). The studies highlight a moderate to high number (47%-92%) of patients would engage in telehealth beyond the Covid-19 pandemic. Fernandes et al. [31] reported 77% of patients selected that they would participate in telehealth, yet the same study highlights 39% feel telehealth is inferior to in-person care.

Clinician experiences with telehealth demonstrated high levels of satisfaction [24,26]. Albahroah & Buabbas [26] suggested that 93.8% of physiotherapists were happy using telehealth as a mode of service delivery with 89% willing to deliver physiotherapy via telehealth. Bennell et al. [24] reported moderate to high effectiveness and satisfaction with telehealth at 70-80% for both 1:1 and class-based services amongst physiotherapists. Physiotherapist's experiences and confidence with using telehealth was high within this study [24], with 70% reporting confidence and positive experiences. Reynolds et al. [25] reported 60% of physiotherapists view telehealth as a sustainable alternative mode of healthcare delivery. However, the same study highlights 40% of physiotherapists feel telehealth is a 'stop gap' during the Covid-19 pandemic and 57% feel telehealth reduces overall job satisfaction, despite 52% reporting an overall positive experience. This sentiment was shared by Fernandes et al. [31] with 55% of clinicians reporting telehealth is not as effective as in-person care and 35% reporting lack of confidence in providing telehealth.

2. *Reduced Travel Time/Reduced Physical Burden*

Several studies identified reduced travel time as a positive trait of telehealth that contributed to satisfaction over traditional healthcare delivery [25,27,28,30]. Reynolds et al. [25] reported reductions in travel time for the service user and was the second highest scoring advantage for telehealth identified by physiotherapists (82.44%). This was also supported by qualitative findings from Barton et al. [30]. The highest scoring advantage was reduced transmission of Covid-19 (92.68%). Odole et al. [28] highlighted the positive nature of telehealth in terms of reducing direct (financial) and indirect (time/physical) costs to the patient when healthcare is not easily accessible. Jansen-Kosterink et al. [27] lists reduced travel time as the initial positive theme expressed by patients within telehealth acceptance and revealed qualitative findings of lowered physical burden for patients as some experienced debilitating exhaustion with travel.

3. *Flexibility*

Two studies identified flexibility of healthcare delivery as a positive theme of telehealth [25,27]. Reynolds et al. [25] reported that flexibility with the delivery of healthcare regarding remote or in-person care for service users, is likely to be one of the most overwhelming positive features offered by telehealth. Jansen-Kosterink et al. [27] identified service user flexibility as its second most listed theme by patients for acceptance of telehealth, highlighting the positive nature this may have for patients who adopt irregular working patterns.

4. *Accessibility*

Several studies mention accessibility as a positive theme of telehealth [24,25,28,30]. Bennell et al. [24] reported that 54% of patients identified ease of access as a positive trait of telehealth; this is further supported by Odole et al. [28] who reported advantages for those who do not have easy access to healthcare or face geographical restrictions. Reynolds et al. [25] showed that 49.27% of physiotherapists reported improved access to healthcare as a positive advantage of telehealth.

5. *Telehealth Vs Face-to-face or Blended Approach*

Bennell et al. [24] showed a moderate number of patients (59%) rated telehealth as the same or better quality as traditional face to face care. Of important note, this statement is made in reference to videoconferencing with 1:1 care, there was a slightly reduced percentage (57%) for class-based care as being similar or better in quality. Malliaras et al. [23] reported a low to moderate (42%) response from allied health clinicians reporting telehealth as equal or superior to traditional face to face care with a low number (25%) reporting that patients valued telehealth to the same degree as traditional care methods. This was further supported by the findings of Fernandes et al. [31] who reported a moderate (55%) to low (39%) number of clinicians and general population stating telehealth as being inferior to in-person care. This notion is further highlighted by Barton et al. [30] with reports that the general population felt that telehealth had overall value but was perceived as inferior to in-person care. Despite this, participants expressed surprise in the value and benefits of telehealth with 3 in 4 agreeing it was good value and financially viable. Large number (85%) of survey participants reported improvements following their telehealth consultation and reported being better or much better (50%) following consultation with participants expressing future telehealth as a supplementary method to in-person care.

Bennell et al. [24] reported a moderate number of patients (47%) were extremely likely to choose telehealth again beyond the Covid-19 pandemic with a low number (28%) reporting they are not likely to engage in telehealth in the future. This statistic was far better for class-based telehealth intervention with a moderate to high number (68%) of patients willing to partake in telehealth in the future and a low number (13%) of patients reporting they would not engage in further telehealth delivery.

Several studies mention the potential for a blended service delivery model [28–30]. Odole et al. [28] highlighted the potential practical implications of applying telehealth to a physical hands-on profession and suggests it can only work as a sustainable model if applied as an adjunct to traditional care methods. Ullah et al. [29] cited similar findings with 69.5% rehabilitation professionals agreeing that both telehealth and traditional community-based services is the best service delivery model. Barton et al. [30] reports several interview participants highlighting a hybrid approach as the best way forward with telehealth provision.

6. *Associations between Variables*

Albahroah & Buabbas [26] reviewed relationships between studied variables amongst physiotherapists and highlighted some insightful findings. Statistical association between participant age (35 – 50 yrs.), professional rank, technology literacy and willingness to engage in telehealth practices were apparent. These findings are not surprising, as experienced physiotherapists may feel more confident with assessment skills and pattern recognition than their less experienced counterparts. The hypothetico-deductive process is likely to be utilised by skilled clinicians and would form part of the well-recognised clinical reasoning process [32].

7. *Lack of Physical Contact*

A negative theme amongst physiotherapists and the general population within several studies was the lack of physical contact with patients from telehealth service delivery [23,24,26,30,31]. Bennell et al. [24] highlighted several issues reported by a moderate number of physiotherapists (n=74) ranging from lack of physical contact to inability to facilitate exercise, thoroughly assess and use hands on treatment techniques. Malliaras et al. [23] and Barton et al. [30] give insight into issues faced by physiotherapists through lack of physical contact; these contributed to impaired clinical reasoning and less certainty establishing the diagnosis. Psychological and physical barriers have been described resulting in over-reliance on subjective

information with limited objective assessment. Physiotherapist's ability to undertake specific physical testing, establish accurate diagnoses and formulate sound treatment plans have been impeded by telehealth. Albahroah & Buabbas [26] declared a lack of therapeutic relationship through non-contact care between patient and clinician as a potential barrier. This was further explored with managers stating issues in detecting physical problems and lack of hands-on interventions and perceived clinical effectiveness as concerns faced with telehealth care.

8. *Computer Literacy/Technology Issues*

Several studies mentioned technology infrastructure as a potential barrier for telehealth service delivery [24,27,31]. Bennell et al. [24] demonstrated a high number of physiotherapists (n=130) reported sub-optimal internet quality and poor computer skills as barriers to engaging in telehealth both with 1:1 care and class-based delivery methods. Jansen-Kosterink et al. [27] identified both technology issues in terms of internet connectivity and lack of computer literacy combined with computer anxiety as barriers to engagement in telehealth amongst patients. This theme was further supported by the findings of Fernandes et al. [31] with digital literacy and access to technology as barriers to good outcomes with telehealth. Alternatively, around half of clinicians (52%) were confident with telehealth and a moderate number (61%) reported as having adequate infrastructure.

9. *Privacy infringements*

A number of publications noted data privacy infringement as a negative feature of telehealth [26,27,29,31]. Ullah et al. [29] reported that around half (52.44%) of rehabilitation professionals had concerns around data security and patient privacy/safeguarding issues during telehealth consultation. This was also supported by the findings of Jansen-Kosterink et al. [27] and Fernandes et al. [31] with patients expressing concern about privacy infringement especially with video consultation methods as the therapist would be able to view their property and surroundings. Albahroah & Buabbas [26] expressed similar findings with a low number of physiotherapists (38%) denoting patient privacy and data confidentiality as a concern with telehealth delivery.

**Risk of bias.** Methodological quality was evaluated using the QuADS (Table 3), CASP (Table 4 and 5), and JBI (Table 6) tools respectively. Each article was reviewed by a single assessor (AS) and verified by a second reviewer (SI).

Limitations in terms of methodological quality were present in several studies and may limit their credibility and capacity to be representative to both patients and physiotherapists. Non-response rates [33] were evident amongst several studies [12,25,26] and may have been enhanced by using web surveys. They have potential to limit responders who have poor internet connectivity or have reduced computer literacy [34] despite lower delivery costs, enhanced design options and less data inputting.

Potential sampling bias in studies either via voluntary response [35] or convenience sampling [36] was evident [12,23–26,30,31] although likely employed due to low cost and time consumption. However, the purposive sampling in qualitative reviews [27,28] had the advantage of selecting information rich subjects to depict important viewpoints on the subject matter [37].

Several studies incorporated a large volume of physiotherapists [12,23,24,31] whilst others had very low numbers [25,29] or did not provide taxonomy of specialities [26,28] limiting adequate representation of MSK physiotherapy views on satisfaction.

Four studies included [24–27] adopted a universal healthcare system with moderate crossover relevance to UK National Health Service (NHS). Socioeconomic inequalities may well exist in countries that endorse self-funded healthcare with the wealthy and highly educated having better access than poorer or less educated populations [38].

**Table 3. Risk of bias (QuADS Criteria).**

| Question | Studies Score (0 – 3) | | | | | |
|---|---|---|---|---|---|---|
| | Bennell et al., [24] | Malliaras, et al., [23] | Reynolds et al., [25] | Miller et al., [12] | Albahrouh et al., [26] | Barton et al., [30] |
| 1. Theoretical or conceptual underpinning to the research | 2 | 2 | 1 | 2 | 2 | 2 |
| 2. Statement of research aim/s | 3 | 3 | 3 | 3 | 3 | 3 |
| 3. Clear description of research setting and target population | 3 | 3 | 3 | 3 | 2 | 1 |
| 4. The study design is appropriate to address the stated research aim/s | 2 | 3 | 3 | 3 | 3 | 3 |
| 5. Appropriate sampling to address the research aim/s | 1 | 1 | 3 | 1 | 1 | 1 |
| 6. Rationale for choice of data collection tool/s | 3 | 3 | 2 | 1 | 3 | 3 |
| 7. The format and content of data collection tool is appropriate to address the stated research aim/s | 2 | 2 | 2 | 2 | 3 | 2 |
| 8. Description of data collection procedure | 2 | 1 | 1 | 0 | 2 | 2 |
| 9. Recruitment data provided | 3 | 3 | 3 | 2 | 2 | 2 |
| 10. Justification for analytic method selected | 2 | 3 | 2 | 2 | 3 | 2 |
| 11. The method of analysis was appropriate to answer the research aim/s | 3 | 3 | 3 | 3 | 3 | 3 |
| 12. Evidence that the research stakeholders have been considered in research design or conduct. | 1 | 1 | 1 | 0 | 1 | 1 |
| 13. Strengths and limitations critically discussed | 2 | 2 | 3 | 2 | 2 | 3 |

Legend: 3 = Excellent, 2 = Good, 1 = Acceptable, 0 = Not evident.

**Table 4. Risk of bias (CASP Qualitative).**

| Questions | Odole et al., [28] | Jansen-Kosterink et al., [27] |
|---|---|---|
| 1. Was there a clear statement of the aims of the research? | Y | Y |
| 2. Is a qualitative methodology appropriate? | Y | Y |
| 3. Was the research design appropriate to address the aims of the research? | Y | Y |
| 4. Was the recruitment strategy appropriate to the aims of the research? | Can't Tell | Y |
| 5. Was the data collected in a way that addressed the research issue? | Y | Y |
| 6. Has the relationship between researcher and participants been adequately considered? | Y | Y |
| 7. Have ethical issues been taken into consideration? | Y | Y |
| 8. Was the data analysis sufficiently rigorous? | Y | Y |
| 9. Is there a clear statement of findings? | Y | Y |
| 10. How valuable is the research? | Include | Include |

Legend: Y = Yes, N = No.

Sample size calculations were absent for all but one study [25] but are important aspects of a study design to draw realistic conclusions from gathered results [39]. Only two studies [23,24] had patient response rates above the threshold set by Krejcie and Morgan [40] to produce a low margin of error of 0.5% and confidence interval of 95%.

Content thematic analysis was used in mixed method and qualitative studies [12,23–28,30,31] allowing for the development of themes extracted from written data and then coded via systematic process [41]. Several studies [23–25,31] analysed textual data from free text answers, whilst others [12,26–28,30] employed the use of interviews and focus group discussion with field notes [12,27], audio recordings [26–28], response validation [26], and

**Table 5. Risk of Bias (CASP Cross-sectional Studies).**

| Questions | Fernandes et al., [31] |
|---|---|
| 1. Did the study address a clearly focused issue? | Y |
| 2. Did the authors use an appropriate method to answer their question? | Y |
| 3. Were the subjects recruited in an acceptable way? | Can't Tell |
| 4. Were the measures accurately measured to reduce bias? | Y |
| 5. Were the data collected in a way that addressed the research issue? | Y |
| 6. Did the study have enough participants to minimise the play of chance? | Can't Tell |
| 7. How are the results presented and what is the main result? | Y |
| 8. Was the data analysis sufficiently rigorous? | Y |
| 9. Is there a clear statement of findings? | Y |
| 10. Can the results be applied to the local population? | Can't Tell |
| 11. How valuable is the research? | Y |

Legend: Y = Yes, N = No.

**Table 6. Risk of bias (JBI).**

| Questions | Ullah et al., [29] |
|---|---|
| 1. Was the sample frame appropriate to address the target population? | Y |
| 2. Were study participants sampled in an appropriate way? | N |
| 3. Was the sample size adequate? | N |
| 4. Were the study subjects and the setting described in detail? | Y |
| 5. Was the data analysis conducted with sufficient coverage of the identified sample? | Unclear |
| 6. Were valid methods used for the identification of the condition? | N/A |
| 7. Was the condition measured in a standard, reliable way for all participants? | Y |
| 8. Was there appropriate statistical analysis? | Y |
| 9. Was the response rate adequate, and if not, was the low response rate managed appropriately? | Unclear |
| Overall appraisal | Include |

Legend: Y = Yes, N = No.

topic guides [26,27,30]. Response validation is a tool used in qualitative research to ensure credibility and rigor of collected data [42] whereas potential subjectivity and unconscious bias may occur from field note taking [43,44].

## Discussion

The studies included within this review depict a predominantly positive impression of telehealth for both patients and clinicians. Findings echo similar studies illustrating comparable positive attitudes to telehealth [10,45–48]. Although a direct comparison between studies was not possible due to their heterogeneity, the overall narrative suggested that both clinicians and patients were happy and willing to engage with telehealth in some capacity.

The suggestion that patients value telehealth to the same degree as traditional face-to-face contact is limited [23,30,31], and there are indications from some studies [25] that this is merely seen as a 'stop gap' measure during the Covid-19 pandemic. However, the Chartered Society of Physiotherapy (CSP) [49] recommends that future services within physiotherapy should adopt a hybrid model of care as this was deemed to be both safe and responsive to patient needs. A wealth of studies [28–31] have endorsed the concept of a hybrid approach to telehealth. However, the decision between in-person assessment and telehealth care should be

balanced against the patient's needs, the nature of the condition, available resources, and the healthcare providers assessment capabilities. Specific criteria for both in-person and telehealth assessment should be decided with appropriate regulatory and policy considerations. The continued expansion of telehealth will require robust policies that address several issues such as financial reimbursement, data privacy and patient safety. Certainly, evidence from this review suggests that policies that promote reimbursement partly between in-person and telehealth care could be critical in sustaining telehealth services for insurance-based models. Flexibility in the delivery of telehealth and its utilisation as a triage tool may offer operational advantages to health care organisations. The ability for clinicians to work remotely or within their own home may help to overcome difficulties with estates and facilities [50] and has the potential to reduce demand for in-person care as demonstrated within other studies [51–54].

The positive themes surrounding patient satisfaction with telehealth, such as increased accessibility, flexibility, reduced travel time and burden, have been illustrated in other studies [10,46,55,56]. Accessibility for patients has often been a focus of telehealth to address some of the health inequalities that exist from patients who reside in remote geographical locations [6,56]. This is likely to translate into other scenarios where access might be limited by physical or financial means. Paradoxically, telehealth may exclude some patients from accessing healthcare if there are difficulties surrounding digital literacy or adequate technology infrastructure [27,31]. Cultural values should also be considered, so that patients from certain vulnerable groups can engage with telehealth in a safe and effective way [57–60]. Factors such as preference of therapist gender and requirements for translator services should be considered when implementing telehealth services within operational and professional policy. A patient-centred approach should be adopted when implementing clinical telehealth services. Considerations of patient convenience, comfort, technological capabilities, need for support, language proficiency and user-friendly platforms must be considered.

Positive clinician satisfaction with telehealth has been demonstrated [24,26] and follows findings from other studies [10,47]. Previous exposure to telehealth increases the likelihood of positive perceptions [61] and developing strategies to facilitate exposure would promote the uptake beyond the pandemic [30]. Broadening experience and problem-solving skills would ensure that future telehealth services provide effective and efficient patient care and should be seen as an additional skill set within the profession [62]. Edifying the value of telehealth in a predominately 'hands-on' profession would be a pivotal part of future provision and policy holders should ensure that stakeholders are engaged in both operational and professional frameworks.

Negative themes highlighted within this review have been previously documented by other studies [6,46,63,64]. Issues such as lack of physical contact, computer literacy and privacy have been acknowledged by the recent CSP recommendations with suggestions that decision making should consider these factors when applying telehealth to patient care [49]. Overcoming professional boundaries is an important part of future provision and this review has highlighted some of the challenges that clinicians face with telehealth. Lack of physical contact and issues surrounding accurate diagnosis have been highlighted by both clinician and patient groups [31]. Clinical scope of practice, safety, and service user preferences should be at the forefront of decision making when implementing telehealth services [65]. Further professional deliberation surrounding risks and benefits of telehealth provision should be considered. Concerns such as communication have been highlighted [30], but risks associated with miscommunication and inaccurate examination and diagnosis should be considered when consenting to telehealth practice [66]. Data security and privacy should be an integral part of operational and professional framework and has been considered in recent core capability frameworks [62].

## Future recommendations

Future longitudinal studies will depict whether clinician and service user views remain positive beyond the pandemic. It could be postulated that positive satisfaction is likely when other services for in-person care were not readily available.

Future research should explore satisfaction of telehealth within condition specific musculoskeletal care and the use of advanced technologies as this may depict a differing view of telehealth for patients and clinicians. For example, the use of advanced technologies may impact and minimise the negative perceptions of telehealth as identified within this critical review. This may also help with implications for clinical practice such a telehealth integration into healthcare services, adopting patient-centred telehealth and expanding regulatory and policy considerations.

Addressing methodological weakness within studies such as non-response bias would seem a logical option in view of some of the low response rates received in several studies [12,25,26]. Future studies would benefit from collection of data from non-responders which all studies failed to achieve. Even with high numbers of responders, non-response bias could still occur if sociodemographic details are inherently different between groups.

Albahroah & Buabbas [26] produced some interesting findings with statistical cross tabulation of variables such as significant associations between age and willingness to use telehealth. Further studies may also benefit from the use of statistical analysis of studied variables as they may highlight correlation between variables that may not necessarily be evident from descriptive statistics and narrative themes alone.

Recommendations for improving external validity of future studies include the engagement of stakeholders in research design and ensuring a disaggregation of physiotherapy specialities. Engaging stakeholders from the outset of research conception will ensure that maximal impact is achieved, and that policy makers and health care institutions are able to appropriately apply and transfer research findings into a clinical context [67]. Disaggregation of physiotherapy specialities will aid future studies in ability to generalise to specific contexts of physiotherapy as patient population and physiotherapy needs are likely to differ considerable within each of these specialities.

## Strengths and limitations of this study

This review highlights and supports the use of telehealth in physiotherapy services and has provided insightful implications for clinical practice and future regulatory and policy considerations. However, it is fundamental to note that this review has explored values associated with satisfaction of telehealth and this should not be confused with efficacy. This review considered the satisfaction of telehealth across a diverse range of patient and clinician populations and a range of varying demographics, age, and geographical locations allowed a broader view of satisfaction of telehealth.

Acknowledgement of the role and contribution of the authors professional background within physiotherapy and extensive clinical and academic experience has enabled a robust understanding of the research topic, highlighting current healthcare contemporary issues.

Limitations exist in drawing firm conclusions on the satisfaction of telehealth between studies due to heterogeneity in patient/clinician populations, healthcare settings, and telehealth technologies used. Several studies had methodological limitations which limit their ability to generalise results. Although this review portrays a positive narrative of telehealth, findings should be applied cautiously to the wider population. This review provided a content thematic analysis of findings, no statistical analysis was undertaken to identify significance between studies.

## Conclusion

This review has partially addressed and answered the research question and study's aims. The research question was to examine the impact of telehealth on patient and clinician satisfaction within MSK physiotherapy. This review has identified literature that supports the use of telehealth within physiotherapy and has culminated a vast number of perspectives supporting the notion of satisfaction from both clinician and patient's viewpoints. The aims of this study were to understand patient and clinician views of telehealth adoption and has revealed some important benefits and challenges faced by current literature, providing a strong basis for future recommendations of research. Although support for telehealth in terms of satisfaction has been identified, methodological weaknesses within studies means that results should be applied cautiously. The findings of this review are supported by findings from the National Evaluation of Remote Physiotherapy Services in 2020 [49] and has been successful in understanding and evaluating patients and clinician's views on telehealth and its adoption during the pandemic. The inclusion of rich qualitative data through mixed method and qualitative study design has provided an additional facet of awareness and deeper insight into both positive and negative challenges faced by clinicians and patients. It also establishes a strong foundation for further research exploring the most suitable services, patient populations, and operational elements for telehealth engagement in MSK physiotherapy.

## Supporting information

**S1 Checklist. PRISMA Checklist.**
(DOCX)

**S1 Panel. Search Terms.**
(DOCX)

**S2 Panel. Search Strategy and Results.**
(DOCX)

**S1 Table. List of Excluded Studies.**
(DOCX)

## Author contributions

**Conceptualization:** Anthony Smith.

**Data curation:** Sue Innes.

**Formal analysis:** Anthony Smith, Sue Innes.

**Methodology:** Anthony Smith, Sue Innes.

**Supervision:** Sue Innes.

**Validation:** Sue Innes.

**Writing – original draft:** Anthony Smith.

**Writing – review & editing:** Sue Innes.

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
