## [Decision Letter · Decision Letter 0]

11 Nov 2024

PDIG-D-24-00415Patient and clinician perceptions of telehealth in musculoskeletal physiotherapy - A systematic review of the evidence-basePLOS Digital Health Dear Dr. Smith, Thank you for submitting your manuscript to PLOS Digital Health. After careful consideration, we feel that it has merit but does not fully meet PLOS Digital Health's publication criteria as it currently stands. Therefore, we invite you to submit a revised version of the manuscript that addresses the points raised during the review process. Please submit your revised manuscript within 60 days Jan 10 2025 11:59PM. If you will need more time than this to complete your revisions, please reply to this message or contact the journal office at digitalhealth@plos.org. Please include the following items when submitting your revised manuscript:* A rebuttal letter that responds to each point raised by the editor and reviewer(s). You should upload this letter as a separate file labeled 'Response to Reviewers '. This file does not need to include responses to any formatting updates and technical items listed in the 'Journal Requirements' section below.* A marked-up copy of your manuscript that highlights changes made to the original version. You should upload this as a separate file labeled 'Revised Manuscript with Track Changes '.* An unmarked version of your revised paper without tracked changes. You should upload this as a separate file labeled 'Manuscript '. If you would like to make changes to your financial disclosure, competing interests statement, or data availability statement, please make these updates within the submission form at the time of resubmission. Guidelines for resubmitting your figure files are available below the reviewer comments at the end of this letter. We look forward to receiving your revised manuscript. Kind regards, Haleh AyatollahiSection EditorPLOS Digital Health Leo Anthony CeliEditor-in-ChiefPLOS Digital Healthorcid.org/0000-0001-6712-6626 **Journal Requirements:**

1. In the online submission form, you indicated that "All relevant data are within the manuscript and its Supporting Information files.". 

3. Uploaded as supplementary information.

2. We have noticed that you have uploaded Supporting Information files, but you have not included a list of legends. Please add a full list of legends for your Supporting Information files after the references list. 

3. We notice that your supplementary files are uploaded with the file type 'Manuscript'. Please amend the file type to 'Supporting Information'. Please ensure that each Supporting Information file has a legend listed in the manuscript after the references list.

4. As required by our policy on Data Availability, please ensure your manuscript or supplementary information includes the following: 

**Additional Editor Comments (if provided):****Reviewers' Comments:** Reviewer's Responses to Questions

**Comments to the Author**

1. Does this manuscript meet PLOS Digital Health’s publication criteria ? Is the manuscript technically sound, and do the data support the conclusions? The manuscript must describe methodologically and ethically rigorous research with conclusions that are appropriately drawn based on the data presented.

Reviewer #1: No

Reviewer #2: Yes

Reviewer #3: Partly

Reviewer #4: Yes

2. Has the statistical analysis been performed appropriately and rigorously?

Reviewer #1: N/A

Reviewer #2: Yes

Reviewer #3: No

Reviewer #4: Yes

3. Have the authors made all data underlying the findings in their manuscript fully available (please refer to the Data Availability Statement at the start of the manuscript PDF file)?

Reviewer #1: No

Reviewer #2: Yes

Reviewer #3: Yes

Reviewer #4: Yes

4. Is the manuscript presented in an intelligible fashion and written in standard English?

Reviewer #1: No

Reviewer #2: Yes

Reviewer #3: Yes

Reviewer #4: Yes

5. Review Comments to the Author

Reviewer #1: the paper has fundamental shortcomings and lacks quality writing, and innovation. However, recommendations for improvement have been provided. Best wishes

Abstract:

•Clarify the specific types of musculoskeletal conditions addressed in the reviewed studies.

•Provide more precise details on the databases searched and date ranges.

•Include the specific number of studies reviewed after screening, not just the initial search results.

•Elaborate on the key findings regarding patient and clinician satisfaction, specifying percentages or ranges where possible.

•Mention any limitations or gaps identified in the current literature on this topic.

•Strengthen the conclusion by highlighting specific implications for practice and future research directions.

Background:

•Provide a more comprehensive overview of telehealth adoption in physiotherapy prior to COVID-19.

•Include statistics on the increase in telehealth usage specifically for musculoskeletal physiotherapy during the pandemic.

•Discuss existing systematic reviews or meta-analyses on telehealth in physiotherapy to better contextualize this review.

•Elaborate on the potential barriers to telehealth implementation in physiotherapy, including technological, clinical, and patient-related factors.

•Clarify the specific gap in knowledge this review aims to address regarding patient and clinician satisfaction with telehealth.

Methods:

•Justify the choice of the PEO format for developing the research question.

•Provide more details on the search strategy, including the full list of search terms and how they were combined.

•Explain the rationale for excluding grey literature and limiting the search to 2019-2024.

•Describe the process for resolving disagreements between reviewers in more detail.

•Include information on how the quality of the included studies was assessed, specifying the tools used and the criteria for evaluation.

Results:

•Present the results in a more structured manner, possibly using subheadings for different aspects of satisfaction (e.g., overall satisfaction, specific advantages, challenges).

•Include a table summarizing the key characteristics and findings of each included study.

•Provide more quantitative data where available, such as ranges or weighted averages of satisfaction scores across studies.

•Elaborate on any discrepancies or contradictions found between different studies' findings.

•Include a more detailed quality assessment of the included studies, highlighting specific strengths and weaknesses.

Discussion:

•Provide a more nuanced interpretation of the findings, considering the heterogeneity of the included studies.

•Discuss how the findings compare to previous reviews or meta-analyses on telehealth in physiotherapy.

•Elaborate on the implications of these findings for clinical practice, health policy, and future research.

•Address the limitations of the review more comprehensively, including potential biases in the included studies and in the review process itself.

•Provide more specific recommendations for future research, identifying key areas where knowledge gaps remain.

Reviewer #2: - Add in the research objectives and research questions as in SLR in Introduction

- Add in the reviewer background (AS) and (SI) such how many years experience

- Highlight in figure 1 - other source from where?

Reviewer #3: The authors examined the impact of telehealth on patient and clinician satisfaction within physiotherapy, and provided insightful recommendations to improve future quality of research in this field. However, this work still contains some inadequacies in the analyses and discussions. Although the authors vigorously reviewed previous studies and summarized them, they only characterized the findings to date. They need to add more specific analyses or discussion to the current manuscript. I listed some issues that need to be addressed in order to improve this work mature enough for publication in PLOS Digital Health.

1.Although the authors present the characteristics of the studies in Table2, the clinical profiles of patients such as osteoarthritis, rheumatoid arthritis, cerebrovascular disease, spinal canal stenosis, etc. are lacking. If the studies the authors referred to include information on patient profiles, they should conduct a deeper analysis of the positive and negative themes of telehealth based on the patients’ profiles.

2.In the “3. Flexibility” subsection of the Result section, the authors explored flexibility as a positive theme of telehealth. However, it is unclear what the flexibility in telehealth means. The authors need to provide a clear definition of flexibility in telehealth.

3.What types of physiotherapy tend to be accepted as telehealth, and what types tend not to be accepted? The authors need to explore the details of physiotherapy menu and provide the tendency in the “7. Lack of Physical Contact” subsection. In addition, the authors should discuss more specifically the possibility of a hybrid approach in terms of the type of physiotherapy.

4.If the authors intend to discuss the future integration of telehealth within routine clinical practice, they should be more specific about how each negative theme could be addressed. For example, with the lack of physical contact, are there digital technologies that can solve this problem? Could virtual or augmented reality (VR or AR) as telerehabilitation improve the negative perception of the lack of physical contact? The authors should discuss the potential of emerging digital technology to address the current problems.

Reviewer #4: Overall, I think this is an excellent review. The discussion is very well done. I really liked the recognition of a paradoxical benefit/harm with technological literacy and cost of travel for patients. The suggestion to have future studies investigating perception after the pandemic vs. intra pandemic is great, was hoping this would be discussed. See below for comments and suggestions.

Would be more clear on physiotherapy - I believe the authors are primarily talking about physical therapy the discipline, but would at least clarify that there are other disciplines (occupational and speech therapy) that are often performed in concert and linked. Are these less amenable or more amenable to telerehabilitation services? Is there a reason why these disciplines were not included in this review?

I think the lack of physical contact should be more discussed. This is a significant limitation in the ability to provide appropriate and corrective physiotherapy. If you are unable to be hands on and training appropriate movement, bad habits can be entrained, which can result in injuries and lack of progress.

Does having physiotherapy as part of primary care make sense? Efficient use of those services frequently involves physiatrists (PM&R) providing a MSK diagnosis or focus for physiotherapists to target

6. PLOS authors have the option to publish the peer review history of their article (what does this mean? ). If published, this will include your full peer review and any attached files.

**Do you want your identity to be public for this peer review?** For information about this choice, including consent withdrawal, please see our Privacy Policy .

Reviewer #1: No

Reviewer #2: No

Reviewer #3: No

Reviewer #4: **Yes: ** James R Devanney

---

## [Decision Letter · Decision Letter 1]

16 Feb 2025

Patient and clinician perceptions of telehealth in musculoskeletal physiotherapy services - A systematic review of the evidence-base

PDIG-D-24-00415R1

Dear Mr Smith,

We are pleased to inform you that your manuscript 'Patient and clinician perceptions of telehealth in musculoskeletal physiotherapy services - A systematic review of the evidence-base' has been provisionally accepted for publication in PLOS Digital Health.

Best regards,

Haleh Ayatollahi

Section Editor

PLOS Digital Health

**Additional Editor Comments (if provided):**

**Reviewer Comments (if any, and for reference):**

Reviewer's Responses to Questions

**Comments to the Author**

1. If the authors have adequately addressed your comments raised in a previous round of review and you feel that this manuscript is now acceptable for publication, you may indicate that here to bypass the “Comments to the Author” section, enter your conflict of interest statement in the “Confidential to Editor” section, and submit your "Accept" recommendation.

Reviewer #1: All comments have been addressed

Reviewer #3: All comments have been addressed

2. Does this manuscript meet PLOS Digital Health’s publication criteria ? Is the manuscript technically sound, and do the data support the conclusions? The manuscript must describe methodologically and ethically rigorous research with conclusions that are appropriately drawn based on the data presented.

Reviewer #1: (No Response)

Reviewer #3: Yes

3. Has the statistical analysis been performed appropriately and rigorously?

Reviewer #1: Yes

Reviewer #3: Yes

4. Have the authors made all data underlying the findings in their manuscript fully available (please refer to the Data Availability Statement at the start of the manuscript PDF file)?

Reviewer #1: (No Response)

Reviewer #3: Yes

5. Is the manuscript presented in an intelligible fashion and written in standard English?

PLOS Digital Health does not copyedit accepted manuscripts, so the language in submitted articles must be clear, correct, and unambiguous. Any typographical or grammatical errors should be corrected at revision, so please note any specific errors here.

Reviewer #1: (No Response)

Reviewer #3: Yes

6. Review Comments to the Author

Please use the space provided to explain your answers to the questions above. You may also include additional comments for the author, including concerns about dual publication, research ethics, or publication ethics. (Please upload your review as an attachment if it exceeds 20,000 characters)

Reviewer #1: I am pleased to confirm that all revisions have been meticulously implemented, and the manuscript is now deemed acceptable for publication. I sincerely appreciate your valuable efforts and the collaborative approach you demonstrated throughout the review process, which has significantly enhanced the quality of this work. I wish you continued success in your research and writing endeavors.

Reviewer #3: The authors have addressed all the comments, and I have no further comments at this time. I believe their manuscript has been improved to a level suitable for publication in PLOS Digital Health. I hope this systematic review will contribute to the further advancement of the field of telehealth.

7. PLOS authors have the option to publish the peer review history of their article (what does this mean? ). If published, this will include your full peer review and any attached files.

**Do you want your identity to be public for this peer review?** For information about this choice, including consent withdrawal, please see our Privacy Policy .

Reviewer #1: **Yes: ** Roghieh Nooripour

Reviewer #3: No
